# Enriching Social Entrepreneurship from the Perspective of Catholic Social Teaching

**John F. McVea \* and Michael J. Naughton**

Schulze School of Entrepreneurship and Department of Catholic Studies, University of St. Thomas, St. Paul, MN 55105, USA; mjnaughton@stthomas.edu
\* Correspondence: johnmcvea@stthomas.edu

**Abstract:** In this paper, we propose that unreflective use of the term social entrepreneurship may perpetuate the idea that "entrepreneurship" is largely a financial and private reality and that this view of entrepreneurship will eventually trivialize or perhaps undermine the important benefits and the real intentions behind the social entrepreneurship movement. We believe that Catholic Social Teaching can shed important light on this dilemma by emphasizing three specific strategies inherent to entrepreneurship when assessing the moral contribution of the firm. As a result, we argue for the principles of *good goods*, *good work* and *good wealth* as an alternative framework for all good entrepreneurial venture.

**Keywords:** social entrepreneurship; catholic social teaching; common good

## 1. Introduction

Despite a lack of consensus over its precise definition (Kannampuzha and Hockerts 2019; Short et al. 2009; Peredo and McLean 2006), it has been widely observed that the term social entrepreneurship has experienced a significant growth in its influence over that last decade (Rey-Martí et al. 2016; Bloom and Smith 2010; Bornstein 2004; Harding and Cowling 2006). A wide range of indicators confirm this expansion in entrepreneurial impact beyond the traditional for-profit world: the rate of new non-profit formation has outstripped that of traditional business formation (Austin et al. 2006a); non-profits contributed an estimated $1.1 trillion to the US economy in 2016, comprising 5.6 percent of the country's gross domestic product (Bureau of Economic Analysis 2020); best-selling books lists include numerous accounts of entrepreneurial successes in the social sector (Counts 2019; Harrison 2018; Patrick 2019; Sud et al. 2009). Alongside these broad phenomena, there has also been notable growth in the academic study of social entrepreneurship (Hota et al. 2019; McQuilten 2017; Zahra et al. 2014; Short et al. 2009). Indeed, there was a five-fold increase in academic publications on social enterprise between 2008–2018 (Gupta et al. 2020). This work has been categorized into a number of important themes and prioritized into the research streams with the most significant potential (Ibid). However, despite this progress and attention, there have been persistent observations that the practice of social entrepreneurship continues to outpace its theoretical development (Murphy and Coombes 2009; Austin et al. 2006b). Nevertheless, there is widespread agreement that the term social entrepreneurship represents an important, perhaps pre-paradigmatic (Nicholls 2010; Stephan et al. 2015), phenomenon worthy of both study and encouragement (Austin et al. 2006b; Hockerts 2017; Weerawardena and Mort 2006).

While in this paper we focus on social entrepreneurship, the spread of this term is part of a broader proliferation of modified entrepreneurial references across the academy. Political entrepreneurship has been used to describe the role of government actors who "coercively obtained resources" for the state (McCaffrey and Salerno 2011). Environmental entrepreneurship has been defined as the use of both commercial and ecological logic to

address environmental degradation through the creation of financially profitable organizations, products, services, and markets. (York et al. 2016). Religious entrepreneurship has been described as "religiously-inspired action for social change" (Adi 2017). There are many praiseworthy aspects to the adoption of an entrepreneurial lexicon, especially to encourage innovation in areas of stagnation. However, we also worry that by doing so we may unintentionally smuggle in some important normative assumptions which deserve a deeper consideration. With this caveat in mind, we will focus the rest of this paper solely on the implication of the wider use of the term social entrepreneurship.

Social entrepreneurship was first used as an expression two decades ago as "one species in the genus entrepreneur. They are entrepreneurs with a social mission" (Dees 1998). Since then, much of the literature has focused on defining the term more precisely. In one study researchers identified 37 different definitions of the phrase in a paper they aptly named "Social Entrepreneurship: Why We Don't Need a New Theory and How We Move Forward from Here" (Dacin et al. 2016). While some have defined social entrepreneurship by the personal characteristics of the entrepreneur, others have used the business sector in which they operate, while others have used their structure (for example, tax exemption). After an exhaustive review of this work, some have concluded that "the definition that holds the most potential for building a unique understanding of social entrepreneurship and developing actionable implications is one that focuses on the social value creation mission" (Ibid). From our perspective, the two most significant contributions of the term have been to encourage a blurring of the line between the for-profit and non-profit sectors, and a re-emphasis on the social impact of organizations.

Social entrepreneurship has been proposed to have several advantages: promoting innovation within non-profits (Dawson and Daniel 2010; Stecker 2014), leveraging and focusing scarce philanthropic resources within communities (Khavul and Bruton 2013; Rahdari et al. 2016), improving access to skilled personnel (Van Slyke and Newman 2006), empowerment of the poor (Yunus and Jolis 2003), faster strategic response to environmental changes (Dees and Anderson 2003), the infusing valuable business skills and tools into the non-business world (Dart 2004), and encouragement for the use of individual creativity (as opposed to government action) to solve social problems (Martin and Osberg 2015; Le Grand 2003). Perhaps most significantly, interest in social entrepreneurship has also shed light on a blurring of the lines between the non-profit and for-profit worlds (Dees 1998; Biggs et al. 2010; Sharir and Lerner 2006; Urbano et al. 2010). As a result, a whole new range of hybrid organizational forms has flourished, focused on solving social problems (Austin et al. 2006a; Alter 2007). For example, in recent years we have seen the creation of new organizations such as hybrid for-profit/non-profits organizations, Benefit Corporations and Low-profit Limited Liability Companies (L3Cs).

However, in addition to the advantages listed above, we believe that the unchallenged addition of the adjective "social" to the term entrepreneurship may also mask some important dangers. Nevertheless, we would like to propose that these dangers may be averted if, instead, we enrich our conceptions of entrepreneurship to incorporate perspectives that have long been at the center of faith-based views of enterprise, in particular, the Catholic conception of the purpose of business. Some critics have highlighted limitations to the scalability of social entrepreneurial solutions (Sud et al. 2009). Still, more have criticized the field for its continued inability to agree upon a robust definition of the boundaries and distinctive characteristics of social entrepreneurship (Kannampuzha and Hockerts 2019; Short et al. 2009; Nicholls 2010; Light 2008; Perrini and Vurro 2006). In this paper, we propose a more fundamental concern. We argue that the most challenging questions are not whether social entrepreneurship is necessary or could be sufficient to solve all the world's problems, nor whether it has not yet been precisely enough defined or described. Our thesis is the following: *the unreflective use of the term social entrepreneurship may perpetuate the idea that "entrepreneurship" is largely a financial and private reality and that this view of entrepreneurship will eventually trivialize and perhaps undermine the important benefits of the*

*real intentions behind social entrepreneurship.* Our thesis is illuminated by the following important questions:

- By encouraging activities and assembling resources to support the 'new' field of social entrepreneurship, might we unintentionally abandon the bailiwick of traditional entrepreneurship solely for private and financial gain?
- Is "social entrepreneurship" seen as a redemptive luxury to be completed independently after the prior maximizing private gains?
- If, the adjective "social" is attributed to a set of activities to describe this particular subset of entrepreneurial activities, does this imply that the rest of the field of entrepreneurship is "non-social", or worse, "anti-social"?

We further propose that the widespread problem in defining social entrepreneurship is simply a symptom of a failure to take seriously the central normative core of "entrepreneurship": that entrepreneurship should have a human end and that this end should be the common good. As a result, attempts to build definitions of social entrepreneurship on purely descriptive contexts and activities risk dichotomizing the field into social/moral and non-social/amoral entrepreneurship. More critically, they risk undermining our understanding of the meaning of work itself.

This essay lays out two risks we associate with social entrepreneurship that are connected to the normative/spiritual dimension of entrepreneurship and work. We then argue that the principle of the common good can help us to understand the normative character of entrepreneurship more deeply, and thus serve as a foundation for social entrepreneurship. The "common good" as a term has become somewhat ubiquitous—often invoked but rarely defined. In this paper, we will take our definition specifically from the tradition of Catholic social teachings rather than from colloquial parlance.[1] In doing so we hope not to be exclusionary, rather we hope to ground our work on a concrete theoretical and theological construct, which will enable us to reach out to other sympathetic beliefs and traditions. This approach also enables us to avoid the all-too-common business temptation of reducing all principles to their utility calculus. Our understanding of the common good is drawn from a body of work called Catholic Social Teachings that includes papal encyclicals, bishop pastoral letters and conciliar and other official documents from the teaching magisterium of the church. Our definition of the common good comes directly from one of its social documents from the Second Vatican Council—Vatican Council II (1965). The social teachings of the Catholic Church seek to provide an "*accurate formulation* of the results of a careful reflection on the complex realities of human existence . . . in the light of faith and the Church's tradition" (John Paul II 1987, #41). The social teachings draw upon the long and varied tradition of Catholicism as well as from "scientific studies promoted by members of the laity, from the work of Catholic movements and associations, and from the church's practical achievements in the social field" (John Paul II 1991, #4). While the social teachings of the Church serve as an "indispensable and ideal orientation" to the good informed by faith, they do not and cannot detail specific answers to every economic, organizational and political problem (Ibid, #43). Catholic social teachings provide theological and philosophical principles to guide discernment into the complexities of the economic and political world. The actual policies and decisions are the domain of laypeople. Thus, from the perspective of Catholic social teaching, we define entrepreneurship by identifying three constituent goods and how they are ordered and relate to each other. These three

---

[1]　We realize that for some, our move to draw upon a distinctive religious tradition seems unnecessarily sectarian and exclusive. We have three responses to this. First, Catholic social teachings has the most extensive development of the common good, which we believe is the key term to understand social entrepreneurship without relegating entrepreneurship to the bin of materialism. Second, while Catholicism is a theologically grounded religion, it is not a sectarian religion. It joins with others of differing religious and philosophical beliefs to explore social and moral issues. Its discourse on the common good for example does not necessarily require acceptance of its theological grounding. Third, there are over 1400 Catholic institutions of higher education globally, many of which have drawn upon social entrepreneurship. If they cannot connect this reality to their specific Catholic mission and identity, there is a good chance that social entrepreneurship will be simply one more academic fad that will give way to increasing budget pressures. For a list of the main documents of Catholic social teachings see https://www.usccb.org/beliefs-and-teachings/what-we-believe/catholic-social-teaching/foundational-documents (accessed on 5 March 2021).

goods will be discussed in more detail below, but for now, we will summarize them in the following way:

Good Goods: making goods which are truly good and services which truly serve.
Good Work: organizing work where employees develop their gifts and talents.
Good Wealth: creating sustainable wealth and distributing it justly (see Dicastery for Promoting Integral Human Development 2018).

## 2. Two Risks of Social Entrepreneurship: Avoiding the Ditches

In reviewing the literature, it becomes clear that there are several dangers, disagreements, and confusions caused by the use of the term social entrepreneurship. We will attempt to disentangle some of these misunderstandings by categorizing the dangers in terms of two major risks. These risks represent the proverbial ditches on either side of the road down which social entrepreneurship is traveling. Every virtue is prone both to excess and defect: social entrepreneurship is no different. As stated above, we believe that we should use the term social entrepreneurship in a manner that reinforces, rather than unconsciously undermines, the social character of all entrepreneurial activities. The risk from the two ditches does not mean we should abandon the idea of social entrepreneurship. On the contrary, it means we should be conscious of the potential and often unintended consequences of both excess and defect. In laying out these risks from the term "social entrepreneurship" we are not advocating a mere name change, but rather that we should be aware of how use of this term could unintentionally negatively affect its own good, as well as the good of entrepreneurship itself.

### 2.1. Defect: The Rhetorical Risk of the Modifier "Social"

There are some rhetorical dangers to an unreflective use of the term "social entrepreneurship". The first risk derives from too narrow or too restricted a use of the term social. In some circles, it has become common to use the term social entrepreneurship as a simple replacement for the term non-profit, in a similar way that non-profit previously replaced the term charity. To some degree, this change could be viewed as a relatively harmless way to make the charitable sector appear more upbeat or au courant. However, we believe that to limit the concept of social entrepreneurship to a rhetorical update would be to sell short its promise in stimulating fundamental new economic activities or in providing a broader field impact for innovation in areas hungry for invention. This promise is nicely summarized in Dees' seminal statement on the meaning of social entrepreneurship:

> "[T]he new name is important in that it implies a blurring of sector boundaries. In addition to innovative not-for-profit ventures, social entrepreneurship can include social purpose business ventures, such as for-profit community development banks, and hybrid organizations mixing not-for-profit and for-profit elements, such as homeless shelters that start businesses to train and employ their residents. The new language helps to broaden the playing field". (Dees 1998)

We agree with Dees that one of the most significant values of social entrepreneurship is its consequent blurring of sector boundaries between for-profit and not-for-profit. We would further argue that the commonly held distinction "between profit-based companies and non-profit organizations can no longer do full justice to reality" (Benedict 2009). The reason it cannot do justice to reality is because the over-emphasis of "profit-status", as a defining characteristic, masks the opportunities for innovation in the structure of organizations. Social entrepreneurship can make a wider contribution by legitimizing the creation of a new "third sector" of the economy (Drucker 1989; Benedict 2009) by illustrating that profit represents only one of many institutional structures which play a role in achieving human and social ends. Benedict explains that "[w]hether such companies distribute dividends or not, whether their juridical structure corresponds to one or other of the established forms [of for-profit or not-for-profit], becomes secondary in relation to their

willingness to view profit as a means of achieving the goal of a more humane market and society" (Ibid).

The phenomenon of social entrepreneurship is an illustration to all entrepreneurs that profit should be seen as a means, not as an end. If, however, this point is ignored, the rhetorical consequences of anointing a subset of entrepreneurial ventures as "social" maybe both significant and detrimental. The obvious implication is that other forms of entrepreneurship must be somehow "asocial" or value-neutral or even "anti-social". Adding "social" to social entrepreneurship can be seen as an honorific term to special organizations who "go beyond the call of duty" within our economy. Widespread use of this "value-added" logic risks a real, as opposed to a rhetorical, division between social entrepreneurship—seen as social and ethical—and entrepreneurship—seen as private and financially driven. This interpretation could reinforce an already significant problem in business, that the sole end of economic trade is profit and that all other prerequisites and consequences are simply instruments towards this end.

The danger of a bifurcation between social entrepreneurship and 'commercial' entrepreneurship is that we create yet one more compartmentalization, a division that distorts the meaning of the work and the role it plays in our lives. We believe it is important at this transformational time in our economy, that we avoid reinforcing or acquiescing to further fragmentation and instrumentalization of our lives. Rather we should confront these forces and provide more integrative frameworks in lieu.

A divergence between "social entrepreneurship" and "entrepreneurship" unwittingly lends support to what some have called the "divided life". This split between the moral, religious life and the professional life has been identified as one of the most serious errors of age (John Paul II 1987). From a more secular perspective, Freeman has described this flawed approach to work and ethics as the "Separation Thesis" (Freeman and Gilbert 1992) which proposes that "work" and "ethics" should be dealt with as conceptually separate. However, from the perspective of social entrepreneurship, we worry additionally about a lesser form of this separation thesis: a sort of "Concentration Thesis" that suggests that the moral responsibility of the business sector can be concentrated in a subset of organizations called social enterprises. This would, presumably, leave the 'regular' enterprise to focus on serving themselves. It would be dangerous indeed to encourage the belief that social entrepreneurship can become the vessel that contains the entire moral contribution of business, relieving conventional entrepreneurship of the troublesome reflections that ethical business requires. These dangers become particularly apparent when we review the literature, which has wrestled for many years with narrowly defining social entrepreneurship as a distinct area of inquiry.

### 2.2. Excess: The Risk of a Boundarylessness

At the opposite end of the spectrum, we believe that there are also dangers from using the term social entrepreneurship too broadly. One of the attractions of social entrepreneurship is the ability of the term to cross the traditional boundaries between the spheres of business, charity, and government. However, there are also risks in unreflectively allowing the term social entrepreneurship to mean almost anything. There are certainly attractive possibilities to viewing a broad array of institutions through the same lens in the hope that non-profits might benefit from some of the discipline and techniques of the for-profit world, or that entrepreneurial firms might be more likely to see profit as a means not an end.

However, within the traditional spheres of government, business and charity there can be important differences of kind and important embedded assumptions that might be overlooked or underestimated by unreflectively applying a universal lens of social entrepreneurship. In this regard, perhaps Brenkert has carried out the most relevant analysis in his critique of the call for a fully entrepreneurial society. He explains that "[t]oo many E-theorists have confused a society in which entrepreneurship is encouraged, with a society in which being an entrepreneur is the overriding goal" (Brenkert 2002, p. 23). Those who believe that entrepreneurship and enterprise can solve our social problems share the

greedy reductionist mindset of the Marxist who believes politics and class struggle are the only solutions to all life's ills.

The application of the entrepreneurship-everywhere mindset risks a conceptual shift that removes the boundaries completely and places no limitations on entrepreneurial action whatsoever. Brenkert explains that "[t]he conceptual extension takes place on a daily basis in a variety of ways. It occurs when people speak or write, for example, about public entrepreneurs, social entrepreneurs, and indeed, the government or academic institutions becoming more entrepreneurial" (Ibid, p. 24). While entrepreneurial qualities are applicable to all organizations (initiative, hard work, creativity, etc.), the "customer relationship" that dominates the relationship between people within the commercial sphere is a different kind of relationship than that which operates between "patients", "students", "clients", or "citizens". Viewing these relationships within the different spheres of business, education, medicine, or the government implies a different set of duties, responsibilities, and freedoms. Brenkart concludes that the advocation of an entrepreneurial society "may become a Trojan Horse whereby various entrepreneurial protagonists gain practical entry for their market values and assumptions into other areas of life. And concerning this move I believe we should be particularly wary" (Ibid, p. 25). This is particularly true when the assumption is that entrepreneurship has only a financial end.

We believe that what is needed is not to return to an inflexible and artificial division between the commercial and the altruistic, or between the public and the private spheres of life. Rather, what is needed is the development of a set of principles that promote and restrain the appropriate role of entrepreneurship in a way that enriches society. "Instead of a dichotomy of public and private, it is clear that we need to think in terms of a continuum structured by various rights and responsibilities. The center of this continuum has greatly expanded with new kinds of organizational response to needs and problems in society that have created, in turn, new overlapping forms of the public and the private" (Ibid., p. 28).

We propose that these two risks of defect and excess, stem from the inadequate normative framework at the core of entrepreneurial studies. As a result, we believe incorporating the concept of the common good can erect valuable guard rails from these ditches. It would achieve this by explicitly defining the "institutional goods" of entrepreneurship that guide an understanding of the good entrepreneur. This principle would also help to order such goods for the integral development of people associated with entrepreneurship. We believe that placing the common good as the first principle of entrepreneurship can help us shed light on specific strategies that underpin what is being called social entrepreneurship in a way that enriches the field without degrading the status of more conventional forms of entrepreneurship.

### 3. The Common Good as a Normative Core Centering Entrepreneurship

From the perspective of Catholic Social Teaching [CST] the Common Good [CG] is not an extrinsic principle imposed upon business, rather it is an intrinsic principle that describes the good which business does, and how this good is related to the development of the people who carry out the work therein. Just as the principles of thermodynamics help us understand what actions will cause a rocket to successfully take flight, so the principle of the CG helps us understand what actions can lead a good business to flourish. The CST tradition contains an extensive and nuanced reflection on the CG which we believe can shed important light on the issues we have developed in this paper so far. It describes the principle as "the sum total of social conditions which allow people, either as groups or as individuals, to reach their fulfillment more fully and more easily" (John 1961, p. 65).

To unpack this description, and to see its relevance for our question, we will start with the very first words: "the sum total of social conditions". It takes many institutions in good relationships with each other to foster the CG. It should be emphasized that no one institution, including the state or business, can embody the fullness of the CG. Rather we should think of each institution as playing a distinctive role in fostering the CG. This is one of the reasons why it is important to identify the risk of boundarylessness from the

term social entrepreneurship. We need a host of institutions, especially the family and religion, but also business, charity, education, health care, as well as the state to pursue the CG. It should be clear that if a society does not have rich, vibrant, and varied institutions then eventually the conditions for a social living will suffer, regardless of the state of the enterprise. Without stable and vibrant marriages and families; a strong, honest, and limited state; an education system that fosters the love of learning, etc., entrepreneurship will not have the capacity to solve our problems. Indeed, in isolation, if forced to carry all our hopes and needs, entrepreneurship will only deepen the challenges we face.

It has increasingly become clear over the last decades, despite the obvious gains from technology, that entrepreneurial culture and the logic of the market, have played a role in commoditizing aspects of our culture, especially concerning education, health care, and charity work. Additionally, yet, without a dynamic entrepreneurial economy, societies would eventually stagnate, and tax-funded government programs would wither. Countries, particularly poorer ones, "that do not have enough business activity tend to lose their best-trained people to other countries because they cannot see a future for themselves or their families in their present situations" (Dicastery for Promoting Integral Human Development 2018, #37). This is why we also focus on the defective risk of relegating entrepreneurship to merely financial or private gain. Such a narrow view of entrepreneurship fails to recognize its specific contribution to the common good.

We are proposing that from the CST perspective we could enrich our conception of entrepreneurship into one of good entrepreneurship. This immediately raises the question "What is the good that entrepreneurship does?" We believe this is a question all entrepreneurs should ask themselves and that if profit is the only answer, then that good is deficient. It would be akin to arguing that the purpose of life is to make blood. Blood is indeed critical to life. For sure, when facing certain ailments, the making of red blood cells can occasionally become an overriding necessity. However, who would argue that the maximizing-of-blood-cells could ever be the actual purpose of life? We have a broader and more multifaceted purpose. Similarly, we would argue that an enterprise, when operating well, creates three interdependent sets of goods that give rich dimensions to the CG of entrepreneurship: *good goods*, *good work*, and *good wealth*. When all three of these goods are present, enterprises contribute positively to the CG by creating the social conditions that increase the probability people will develop. Behind each of these interlocking goods are principles that help the leader to order these goods to create a community of persons that serves to help people who are connected to the entrepreneurial firm to flourish (see Dicastery for Promoting Integral Human Development 2018; see also Specht and Broholm 2009). Below we describe each of these three goods and link them to their corresponding principles. We will then discuss each good concerning both entrepreneurship and social entrepreneurship.

*3.1. Good Goods*

The first and most central good produced by the good entrepreneur is the creation of goods that are truly good and services that truly serve (Goodpaster 2011). When we speak of *good goods*, the principal stakeholders are suppliers and customers—inputs and outputs. Two principles help us to define the good of the product. The first principle is "the good entrepreneur creates goods and services which meet the needs of the world". Business is the principal way we produce food, shelter, clothing, communications, transportation, medicine (pharmaceuticals, devices, procedures), etc. It is the institution where we often get access to basic products and services. For example, in pursuing efficient methods of production and distribution entrepreneurial businesses have brought down the cost of the basic items we need to liberate us to live rich complex lives beyond subsistence. Companies such as General Mills, Pillsbury, Cargill, Kraft, Target, Supervalue, Monsanto, etc. have enabled us to spend a smaller percentage of our income on food than for any people in the history of humanity. This progress has served everyone, especially the poor and the hungry.

Unfortunately, these benefits can also come with the unintended consequences of bland and unhealthy food, obesity, soil erosion, commoditization of farm labor, significant downward income pressures on the family farm, etc. Every good and service produced can have negative externalities, which businesses need to constantly mitigate and, when possible, eliminate if they are to truly meet the needs of the world. *Good goods* are rarely ever absolutely good and thus businesses must hold constant responsibility to discern and be accountable for the negative effects of their actions, even if they are unintended. When businesses attempt to rationalize or 'greenwash' their negative externalities (pollution, health effects, poor use, etc.) they must be criticized. However, the good that businesses do by creating their core goods and services can be so obvious that it is frequently either misunderstood or overlooked by the same critics.

Two examples highlight this point. The first is an example of over-romanticizing dramatic inventions and failing to recognize the significance of the ordinary in relation to meeting the needs of the world. Reell Precision Manufacturing has developed a torque technology that produces some of the best hinges and clutches in the world. However, one of the founders of this successful business expressed to us that one of his great regrets is that he "did not develop a product that saved lives, such as cardio pacemakers instead of just making simple clutches and hinges". He often wished his product was not so *ordinary*. However, ordinary is derived from the word order. The infrastructure necessary for an ordered society that allows people to develop entails a multitude of products and services, including hinges and clutches, bolts, tables, chairs, steel, toilets, plastic, roads, as well as pacemakers, microfinance, fair trade coffee, social services, affordable housing and so forth. In other words, we should not be overly romantic about *extraordinary* innovations, which dramatically affect but a few, while discounting how the *ordinary* orders our world through the supply of the regular goods and services entrepreneurship provides. It takes a lot of products and services, from the lifesaving to the mundane, to order a good society. As purpose-driven organizations, the creation of these products and services is often the most significant good that the organization creates. Yes, making hinges, as *ordinary* as that may be, can be a morally good enterprise.

This brings us back to the "unintended consequences" of social entrepreneurship. We often think that to be a good business, one either has to focus on a corporate philanthropy program or be involved in some ex-ante "social cause". As a result, simply creating products that work well and services that serve can be seen as merely morally neutral. Thus, the scope of the *good* of entrepreneurship can be diminished into the volunteered time of employees and the charitable contributions the company can afford from its residual profits. However, from the perspective of CST Timberland is not a good entrepreneurial venture just because it uses its profits to support the volunteer organization City Year. It is primarily a good entrepreneurial venture because it makes good boots and shoes. Good boots and shoes are products that meet the needs of the world. They free customers to be more active, to gain access to nature, to work safely on building sites, to help create housing, and to spend less on replacement footwear.

Bringing effective products and services into existence is an essential element of whether an entrepreneurial business is good or not. They can contribute critical goods to the lives of millions. Entrepreneurs who make hinges, plastic molds or boots in ways that foster quality products through endurance, through resisting competitive pressures to produce cheap, inferior, or deceptive products, are "meeting the needs of the world" and contributing to the CG just as those are who make micro-loans to the poor or providing shelter to the homeless. There are no boundaries to the scope of morality and ethics in our lives.

Reinforcing the moral content of 'business as normal' then raises the question of whether all products and services are good. Within today's culture of consumerism, there is often a default belief that "if the market wants it, and law allows it, the product and service must be good". Further, the spirit of entrepreneurship and innovation often encourages the belief that "if it can be done, then it will eventually be done. Nothing is impossible"

This is sometimes followed up by the assertion that "Therefore, it should be done. And done by us. First!" This logic of the market can be used to justify a whole series of legal products and services such as tobacco, weapons, pornography, gambling, genetic therapies, exam cheat web sites, pay-day loans, rent-to-own services, speculative activities, violent video games, pirating music sites, so-called gentleman clubs, and so forth. At the heart of *good goods* is a distinction between needs and wants. Rather than simply asking "Are they legal?" or "Will they have a positive cash flow?" "Will *someone* be willing to buy them?", we need to ask how and to what extent will these products and services serve the needs of the world. From this perspective, it becomes obvious that the nature of the good created by new products and services does not rest on whether the organization is structured as a for-profit, a non-profit, or whether it is given the honorific title of a social venture. Rather, the heart of the question is whether this product or service is produced in a way that best contributes my skills to meet the needs of the world?

Commercial entrepreneurial companies can and do, do good by serving the needs of the world. However, from the perspective of CST, there is a second principle of *good goods*: "the good entrepreneur maintains solidarity with the poor by being alert for opportunities to serve otherwise deprived and underserved populations and people in need". This solidarity often comes from a deep sensitivity to the sufferings of the vulnerable. Solidarity is not just a "feeling of vague compassion or shallow distress at the misfortunes of so many people, both near and far". On the contrary, it is a firm and persevering determination to commit oneself to the common good.

The social entrepreneurial movement has been important in bringing to light many hidden or ignored problems of our capitalistic economy, namely the marginalization of those who cannot afford access to the market because of their limited purchasing power. Further, it has highlighted that human ingenuity can be applied to these problems sometimes in ways that are more effective than traditional governmental or traditional charitable solutions. Thus, there are two important ways social entrepreneurs can seek solidarity with the poor in relation to *good goods:* creating goods and services that serve the poor and creating empowerment and generating respect and self-respect amongst those in need.

Too often markets can overlook the needs of the vulnerable and penalize them through higher prices, rent-to-own schemes, inconvenient locations, unreliable products and services, etc. For example, pawnshops, banks, credit shops, all charge higher interests to the poor, which either prevents them from accessing credit or forces them to spend a higher percentage of their income which decreases their probability for success. In response to this injustice, microcredit organizations such as the Grameen Bank, Accion, Catholic Relief Services, etc. were created to help the poor become protagonists in their development. When affordable and accessible credit is combined with products and services that serve the poor, such as workable and inexpensive water pumps, cookstoves, refrigeration, along with renewable, cheap, energy sources, social entrepreneurs give empowerment to the poor and to addressing the problems and root causes of poverty.

These social entrepreneurs have created new ways of doing business precisely because they were in solidarity with the poor. In contrast, commercial businesses too often have distanced themselves from the poor. Yet, all businesspeople, not just social entrepreneurs should have the poor particularly in mind. This type of work can sensitize companies to the suffering of those in developing world countries, encouraging the development of workable solutions to distribute products to the developing world at a more affordable price.

### 3.2. Good Work

From the perspective of CST, the second good created by the good entrepreneur is organizing *good work* where employees can develop their gifts and talents. Thus, at the heart of *good work* is the principle: "*The good entrepreneur makes a contribution to the community by fostering the dignity of human work and its subjective dimension.*" To understand

the significance of this principle, we need to first grasp what is meant by the "subjective dimension" of work.

When an entrepreneur works, he or she is conscious of achieving goals, often described in terms of productivity, efficiency, profitability, quality, and so forth (Mele 2001). Most entrepreneurial education and literature focus on how to change or maximize these objectives: discern opportunities in market's niches, build business through a relationship with customers and suppliers, manage the day-to-day cash flow to fuel growth, develop a team of employees with complementary skills and talents, and learn to effectively create systems as the business grows. Yet, the work of an entrepreneur, whether social or conventional, does not solely consist of goal achievement, since the activity of work does not only begin and end in the transformation of objects. The activities of workers do not only impact the physical world, but they also impact and change themselves.

As a self-reflexive activity, work reflects right back into the person carrying out that work. When people work, they make decisions about what to produce, that is, the objective dimension of work; however, they also simultaneously make choices that affect themselves, for instance, through their labor, they can either choose to honor their potential or demean themselves (John Paul II 1981). Just as we change the world through work, so does our work change us. This concept is aptly illustrated by the image of a blacksmith in a children's cartoon. Through hammering at the anvil, the smith works the iron transforming it into horseshoes. However, over time the blacksmith has also been transformed through the work. His arms and chest are swollen by the muscles his hammering has built, and his unworked lower body is wasted and tiny by comparison. This image reminds us of the fact that we need to reflect not just on how our work has consequences for the world, but that the work we do transforms us and our character.

The impact of this subjective dimension becomes even more important and troubling when we apply this concept to how we think at work. In professional jobs it is not the repetitive use of our forearms that distorts us, rather it is the unbalanced or repetitive thinking of the lawyer which might diminish their compassion, or the repetitive quantification of the accountant that might overdevelop their economic rationalizations. Despite these dangers, the entrepreneurial literature's focus is on financial success and private aspirations, and there seems to be a certain level of agnosticism toward the subjective dimension of work. The growth of the entrepreneur as a person, or the lack thereof, through the process of entrepreneurship is commonly treated as if it were solely a private issue. In effect, within the academic literature, the entrepreneur herself has been taken out of entrepreneurship. This problem can only be exacerbated if the implication of the term social entrepreneurship is that regular entrepreneurship is private and nonsocial.

At this point, we need to address a kind of poverty rarely mentioned in the social entrepreneurship literature, namely the spiritual poverty that is often found in both the theory and practice of entrepreneurship. This problem centers around the private character of the entrepreneur and how that diminished character could unintentionally be reinforced by extracting "social entrepreneurship" into a separate field of inquiry. This is not to say that social entrepreneurship has created this problem, but rather that it could tacitly bolster the belief that issues of character are best dealt with privately outside the process of entrepreneurship itself.

The privatization of character can be seen in the way we talk about work and in what we do with property that is widely held as a personal and private choice: "So long as it doesn't hurt anyone, I have the right to do whatever I like with what is mine". Underlying this understanding is a form of philosophical individualism that implies at least two things: (a) exclusivity of ownership: "this work and property is mine and thus it is not yours" and (b) control: "since it's mine, I alone determine what it is to be used for within the limits of the law". However, as Augustine pointed out, the word private comes from "privation"—a loss of meaning or substance—that is, spiritual and moral poverty. To understand work and property only in private terms is to refuse to recognize any inherent social and gifted character, and instead reinforce an implied moral relativism in entrepreneurship that

is found in the often-repeated mantra: "Inside every self-made man is a poor kid who followed his dreams". Behind this mantra is a vision that entrepreneurs are special, they can do whatever they dream, they are non-conformists, everything that they create is theirs, *ex nihilo*, they owe no duty to anyone but themselves, they are their own bosses, and they decide what is right or wrong for them. This highly individualistic and relativistic line of thought can only lead to the sort of moral wasteland that makes it impossible to engage entrepreneurs in the spiritual dimensions of their work and property. It also makes it unlikely that entrepreneurs will grow morally and spiritually through their work and create damaging tensions between the person they are when at work and the person they are trying to become in their private life

From the perspective of CST and the subjective dimension of work, the good created by the development of good entrepreneurial character cannot simply rest on whether the organization is structured as a for-profit, a non-profit, or whether it is given the honorific title of a social venture. Mohammad Yunus, Nobel Peace Prize winner and founder of the Grameen Bank, who helped pull millions out of poverty by popularizing micro-loans to women, structured his enterprise as a commercial for-profit bank. Despite for-profit status, his leadership has marked him as one of the great international moral leaders of the last quarter of a century. Equally, most of us have become aware of leaders of charitable non-profit foundations who have allowed workaholism, self-serving, complacency, and other entrepreneurial vices to diminish them as human beings.

Good entrepreneurs recognize, however, that work affects not only their character but also the employees with which they work. From the perspective of CST, this recognition should move good entrepreneurs to structure work according to a second principle of "good work", namely, subsidiarity. This principle fosters the growth of employees by providing opportunities for them to exercise their talents and skills as they contribute to the mission of the organization. The principle of subsidiarity within an organization fosters three responsibilities of both the entrepreneur and the employee.

First is the responsibility to design the work of employees in a way that taps their talents and skills. Howard Rosenbrock, a manufacturing engineer, argues that "[i]f the engineers could think of people as if they were robots, they would give them more human work to do". He explains that too often engineers design work for people using only a fraction of their talents, skills and knowledge. Engineers would never waste the hard-earned capabilities of robots or artificially intelligent systems. They would take care to design the work around the full potential of the robot, otherwise, the engineer would be considered wasteful and inefficient. Rosenbrock explains that most engineers would try to reorganize the production process to use the robot more effectively and extensively by allocating additional tasks in the production process for it to do if it were capable (Rosenbrock [1981] 1985). It would offend their sense of good design practice if the capacities of a highly advanced device were not utilized fully.

The paradox here is striking: engineers and managers often feel no such need to make better use of the available talents, skills and abilities of the human worker. Indeed, human workers are often woefully underutilized and actively de-skilled in order to minimize the compensation they might be due. The principle of subsidiarity requires entrepreneurs to take seriously the task of drawing the full talent, knowledge, abilities and skills out of workers by creating a culture that invites initiative, innovation, creativity, and a sense of shared responsibility. The two rules of thumb that guide subsidiarity is "give as much freedom as possible and as much authority as necessary" and "push down decision making to its most appropriate level".

Second, good entrepreneurs should teach, develop and appropriately equip employees, making sure they have the right tools, training, and experience to carry out their tasks. To define one's work broadly, but not to provide the education and skills to competently carry out such work, is a recipe for failure both personally and organizationally. The general rule of thumb here is that those closest to the work often know the most about the work, especially when well-educated and equipped. This principle requires us to care

for the development and flourishing of the human beings who work for us while they are working, rather than merely hoping that the financial returns of work can allow them to flourish outside of work when they go home.

Third, good entrepreneurs establish strong relationships with their employees when they delegate with deep trust. When entrepreneurs take upon themselves, in full trust, the risks of delegating decision making to lower levels, they are conferring a significant authority upon the employee. In return, when employees exercise this authority in freedom, responsibility and competence, bonds of trust are formed strengthening the relations between them. Taking on the risk of the decisions is what can transform delegation from a mere technique of management to delegation as part of the virtue of trust, which contributes to strengthening the bonds of connection. One who merely delegates to maximize efficiency is one who will take back authority at any time, often at times of highest stress, which is precisely the crucible where the deepest trust can only be earned. In such a situation, employees are not called to the same level of excellence and participation as in a situation governed by the principle of subsidiarity, where delegation is part of a deeper relationship that is being formed. The rule of thumb here is "to avoid micro-management".

These principles, "*dignity of work*" and "*subsidiarity*" should inform the design of the workplace for the good entrepreneur. The danger for entrepreneurs is that they may not see all of this as their business: an idea potentially reinforced by the implication that "if you want that sort of thing then you should go work for a social venture or a charity". In contrast danger for so-called social entrepreneurs is that their definition of the good can become fixated on the *goods or services* they provide, and they can neglectfully fail to create good work. Even altruistic organizations can turn into "sweatshops serving the needy" or compassionate organizations run autocratically "because of the critical needs of our clients". No matter how good their goods, if organizations rob employees of opportunities to exercise personal initiative, lessen their responsibilities, or burn-out critical workers then, no matter the righteousness of their cause, they will produce injustice. An innovative charter school principal focused on pulling its students out of poverty, who forces teachers into repetitive thoughtless learning plans, constrains teaching by over rigorous testing, underpays staff, and allows no voice of dissent is not a good entrepreneur regardless of the percentage of students admitted to college.

*3.3. Good Wealth*

There are two important dimensions of good wealth: creation and distribution. These two elements of value creation are often juxtaposed or separated, however, from the perspective of CST, they cannot be truly understood apart from each other. We cannot distribute wealth if we have created none, nor can we sustainably create wealth without somehow fairly distributing it to those who have created it. These two dimensions need to be understood in a principled way. In terms of wealth creation, entrepreneurial enterprises exercise stewardship over the resources when they create more than what they have been given. Like the good stewards in the Gospel parable of the talents, entrepreneurs produce wealth by creatively utilizing the resources at their disposal and finding innovative ways to produce more than what they have received (Matthew 25: 14–30). In the world of business, this is commonly called "value added". One necessary dimension of this value added is economic, namely profit—the surplus of retained earnings over expenses that enables an enterprise to sustain itself into the future. The good entrepreneur uses resources effectively and maintains adequate levels of revenue, healthy margins, utilization of capacity, strong levels of productivity and efficiency, all in order to ensure the viability of the organization.

When an enterprise makes a profit, it generally means that the factors of production have been properly employed (John Paul II 1991, p. 35). A profitable business, by creating wealth and promoting prosperity, helps individuals excel and realize the common good of society. If profits are not created, they cannot be distributed or reinvested, and organizations die. Profit is like food. You need it to be healthy and sustainable, but you do not live for it. It is a good servant, but it makes a lousy master.

Wealth creation brings with it the concomitant task of wealth distribution. As soon as wealth is created, one must distribute it. The challenge is not in carrying out distribution, but whether that distribution is guided by principles of justice. According to CST the principle of just distribution calls for wealth to be allocated in a way that creates "right relationships" with those who have participated in the creation of such wealth. This principle raises a set of knotty and enduring moral challenges for entrepreneurs, challenges that touch upon fundamental questions of equity and fairness. Among other things, the principle of just distribution calls leaders to discern and account for the moral implications of how they set prices, compensate employees, manage payables and receivables, pay taxes, and allocate benefits and support within their community. All of these issues deserve significant attention, but in light of our theme of social entrepreneurship, we will only focus on the last of these.

As discussed earlier, countless entrepreneurs do not provide a product or service that implicitly brings "social entrepreneurship" to mind. Nevertheless, many still allocate a portion of their profits for distribution to the poor or needy. While we do not want to overlook the obvious fact that entrepreneurial firms do a lot to contribute to poverty eradication by simply generating employment, paying just wages, charging fair prices, paying taxes, etc., entrepreneurs also share additional and particular responsibilities to the poorest of the poor. Need will never be eliminated, even by millions of good entrepreneurs following good missions for particular purposes. There will always be some who fall through the cracks, whose needs are never served, for whom 'natural' economic intervention will come too late. The situation of the haves and have-nots has created a terrible division in our society, and the distance between rich and poor has increased substantially (growing Gini coefficient ratios; segmentation of rich and poor in zip codes; gated communities; etc.). When the poor suffer and the rich are satisfied, there cannot be unity or complacency by anyone. From a CST perspective, because entrepreneurs create wealth, they have a distinct responsibility to the poor, whether they have caused such poverty or not.

The motivation for entrepreneurs to distribute a portion of profits to the poor is often premised on either a religious duty to tithe or on a humanistic impulse to help, or on both. One group of businesses that have done this interestingly is called the Economy of Communion (EOC), approximately 800 businesses world-wide, mostly small to medium-size enterprises, who have committed to distributing their profits in three ways: one part goes to the poor, one part for re-investment back into the company, and the third part for a cultural formation program they have developed (Gold, Markets and Morality). Inspired by the biblical command to tithe, these EOC companies promote the idea that giving to the poor is not a convenient afterthought of the business but is crucial to the self-understanding of the business itself. Profits are not only for shareholders but for the poor.

As before, from the perspective of good entrepreneurship, it becomes obvious that the good created by solidarity with the poor does not rest on whether the organization is structured as a for-profit, a non-profit or whether it is given the honorific title of a social venture. However, all entrepreneurial ventures cannot and should not be expected to be equally altruistic or altruistic in the same way. For example, it is good that a financially stretched substance-abuse venture does not hand out its last $1000 cash to the United Way and, as a result, fail to honor its own payroll. On the other hand, a $1 million cash donation to his or her alma mater will not offset the damage caused by the entrepreneur who runs a polluting factory. Andrew Carnegie gave millions of dollars to social causes, but the relations amongst his employees, suppliers, unions, etc., were debilitating.

The corporate philanthropy of many entrepreneurial firms will always be constrained by the ability of the organization to continue to carry out the other institutional goods of the business. Philanthropy which is funded by the production of stripped-back or shoddy goods, or by mistreated or exploited employees is clearly not serving the interests of the CG. Good entrepreneurial philanthropy should creatively recognize the entrepreneurial constraints that occur across the venture's life cycle. It might make little sense to donate cash to needy causes in the cash-strapped early years and risk the stability of the venture,

but this does not mean that the organization should defer philanthropy. It might mean that this good is better served by altruistically donating employee time, or goods and services, or by sharing expertise with the poor and needy. Equally, in a highly profitable growth stage, it might make more sense to donate a portion of profits to the needy rather than take employees off the job at a critical time. However, none of the altruistic tactics change the fundamental good that responsibility to the neediest in our society is required of the good entrepreneur. These questions can apply to an exploitative non-profit that treats its employees and volunteers without respect, or a for-profit sweatshop, or an AIDS charity bike ride that spends 80% of expenses on riders' expenses and donates less than 5% of its turnover. The nature of the good is determined by the detailed day-to-day actions of the entrepreneur, not by declarative mission statements and/or financial structures.

## 4. Conclusions: Seeing Things Whole

In summary, use of the term social entrepreneurship has been popularized as an important and valuable part of a broadening of entrepreneurial studies over the past two decades. There is much to laud in exploring the study of innovation and change within non-profit sectors and in encouraging innovation in new types of hybrid organizations. However, in this essay, we have argued that unreflective adoption of the term social entrepreneurship may also cause some collateral damage. Too narrow a definition can simply result in a trivial rechristening of charitable or nonprofit sectors. More dangerously unreflective use of the term can also lead to a rhetorical separation between social entrepreneurship (with a normative mission and core) and 'regular' entrepreneurship (now freed of normative values and purpose beyond that of profits). On the other hand, too broad a definition risks turning everything in life that involves change into some form of entrepreneurship. As a result, some normative values, embedded within the traditional theories of entrepreneurship may be thoughtlessly imported into areas of human endeavor. Doing so may also crowd out some important historical normative values.

We believe, rather than widely adopting the term social entrepreneurship, that we would be better served by directing all the energy and inspiration provided by the successes of so-called social entrepreneurs to reflect more deeply on our understanding of the broader impact of entrepreneurship itself. We have argued that all forms of entrepreneurship are founded on normative values, whether these are explicitly acknowledged or not. Furthermore, some of these normative values are better than others. It would be impossible to practice something as pervasive as entrepreneurship without engaging normative values. Moreover, if our 'social' values are restricted to social entrepreneurship, this may, by implication, give free rein for 'regular' entrepreneurship to justify a narrow focus on values such as mere efficiency or financial profit.

On the contrary, we have argued that, as an entrepreneur, I cannot achieve my good except by ordering it toward your good in such a way that together we develop a community where we all develop integrally. This principle should encourage us to reflect deeply on the ethical and moral contributions of all entrepreneurs within our society. We believe that, rather than encouraging the reinforcement of the term social entrepreneurship, which may trivialize or dichotomize the study of value creation and trade, that it would be better to apply the term entrepreneurship to all forms of innovation ventures regardless of their legal structure or the explicitness of their social purpose. This would enable us to apply ourselves to exploring more deeply the multiple principles of good which can be applied to entrepreneurship across a myriad of contexts. Thus, we propose that what is really at stake in entrepreneurship is the nature of the good created, the nature of the work producing it, and the nature of the wealth which results, regardless of how the organization is structured, the name or category we use to categorize it or the espoused righteousness of the values it declares.

We believe that Catholic Social Teaching and the principle of the common good provide an important framework through which this exploration can be carried out. This perspective emphasizes three strategies inherent to entrepreneurship when assessing the

moral contribution of the firm. As a result, we argue for the principles of *good goods*, *good work* and *good wealth* (see Figure 1) as an alternative framework for all good entrepreneurial venture. For example, employees within a firm can only flourish as employees (good work) when they serve the needs of those outside it with good products and services (good goods) and only when they create and distribute wealth (good wealth). These three forms of good, which are the consequences of good entrepreneurship, represent three specific areas of strategy good business may pursue. As a result, we have outlined several specific examples of ways in which good entrepreneurs may contribute to, or neglect, the common good through value creation and trade. No human institution, then, "can escape the issue of its own common good . . . if it is to foster the development of people" (Pontifical Council for Justice and Peace 2005, p. 165). As Thomas Aquinas pointed out 700 years ago: "a man's will be not right in willing a particular good, unless he refers it to the common as an end" ( Aquinas 1947, I-II q. 19, a. 10, Reply).

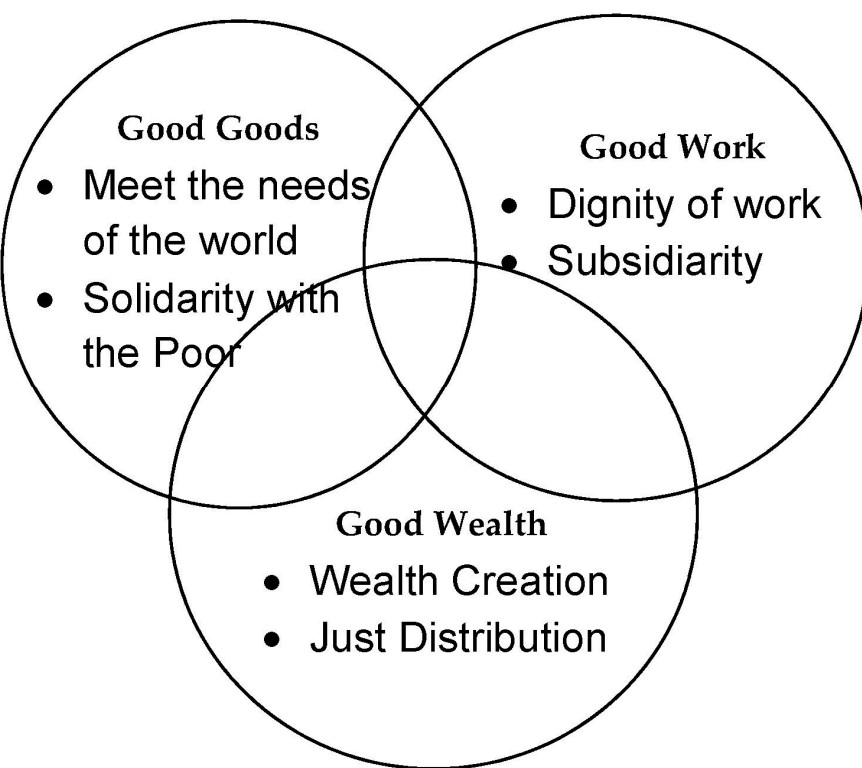

**Figure 1.** The Three Goods of Entrepreneurship.

**Author Contributions:** Both authors contributed to this article. All authors have read and agreed to the published version of the manuscript.

**Funding:** No funding was received for this article.

**Institutional Review Board Statement:** Not applicable.

**Informed Consent Statement:** Not applicable.

**Data Availability Statement:** Not applicable.

**Acknowledgments:** This article was prepared with the help of research assistant Erin O'Connor.

**Conflicts of Interest:** The authors are not aware of any conflicts of interest that might have affected completion of this work.

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
