# Peer review of "Enriching Social Entrepreneurship from the Perspective of Catholic Social Teaching"

_religions, doi:10.3390/rel12030173_

Round 1

Reviewer 1 Report

In this paper, the authors highlight how the unreflective use of the term ‘social entrepreneurship’ may be misleading. They also shed new lights on the meaning and substance of the concept of ‘entrepreneurship’ by drawing wisdom from Catholic social teaching. The essential arguments are insightful and reasonable. The paper is generally well-written, and I believe that it will be a welcoming addition to the increasing discussion on how we should perceive and understand ‘entrepreneurship’. That said, I think the paper will benefit from revision in three areas before publication.

  1. It is commendable that the authors have already engaged with a wide range of literature discussing social entrepreneurship. However, the paper’s central part is more on the concept of ‘entrepreneurship’ rather than the modifier ‘social’, so I think this paper will be improved if the author could reconstruct the introduction in two ways.

      Firstly, the problem of the reflective usage of entrepreneurship exists not only in social entrepreneurship but also in many other emerging fields, including policy entrepreneurship, environmental entrepreneurship, and more recently, religious entrepreneurship. Therefore, I would suggest the authors generally discuss the common themes or problems across these different varieties of entrepreneurship before using social entrepreneurship as a particular example to highlight these common problems. One paragraph at the very beginning of the article will do.

      Secondly, I would suggest the authors discuss the different definitions or perspectives in social entrepreneurship. There have been plenty of reflections on the essential characteristics of social entrepreneurship. Many definitions of social entrepreneurship focus on the innovation and risk-taking new operative paradigms for social benefits. It is fair to give a mention of these existed efforts. Moreover, for readers who may not be entirely familiar with social entrepreneurship, the authors are obligated to explain and elaborate on the very concept before commenting on its advantages and disadvantages.

  1. The exact meaning and scope of Catholic social teaching need to be defined more explicitly in the paper. As far as I am aware, there is no official canon of principles or documents associated with Catholic social teaching, even though the phrase is widely used among modern and contemporary Catholic theologists. Each successor of Pope Leo XIII emphasises slightly different aspects of the term. Therefore, it is inappropriate to assume the concept as given. Instead, the authors need to clarify what they intend to describe or cover by the term. Moreover, Religions is a general journal, so not every reader is familiar with Catholic theology. The authors should also provide some background information on Catholic social teaching to readers who are experts of other religions.
  2. The current conclusion looks a bit too ‘rash’ and ‘thin’ to me. It would be helpful for the authors to summarise their substance and arguments in a more structural and orderly manner. It might also be helpful to discuss how the insights from the Catholic social teaching relevant to people who believe in other religions or do not believe in religion at all. After all, Catholic social teaching has been developed in criticising many other modern ideologies.

Beyond these three major issues, I would suggest the authors address a couple of minor issues. For example, there is no need to place a full stop at the end of the title. Also, some paragraphs are very long, and I would suggest the authors break them down where possible.

Generally speaking, I think this is a very decent article that I enjoy reviewing, and I believe its quality will be further improved after some minor revisions.

Author Response

Dear Editors,

Thank you so for the opportunity to resubmit the article Enriching social entrepreneurship from the perspective of Catholic social teaching”. We would like also to thank you, the Special Editor, the Assistant Editor and the reviewers for providing us with such clear feedback which we used to guide our rewriting of the paper.

As was suggested, we have focused our edits to the paper on the concerns Reviewer 1 and 3 raised with regard to what they felt was too abrupt an ending. As a result, we have completely rewritten the conclusion: summarizing our argument, clarifying our proposition, and specifying how we should better address both the term and the underlying phenomenon, of social entrepreneurship in the future. Second, we have added some foundational details better defining the terms Catholic Social Teaching and the Common Good. We found this suggestion most helpful and agree that it helps to make the article more accessible to those of different backgrounds, which is indeed one of our aims. However, to not distract from the flow of the argument we placed the body of this explanation in an extended footnote, for those with an interest in these details. In response to the more specific items of feedback, we have listed in detail the changes on the following pages.

We are pleased to have had the opportunity to continue to work on this paper to strengthen it and make it of greater value and impact. We greatly appreciate the work of the reviewers in providing direct and useful feedback which encouraged us to make these improvements. Please let me know if there is anything further you need from us. We are excited at the possibility of getting this article published with you. We also look forward to working with you again in the future.

The authors.

1.   Responses to REVIEWERS’ Comments and Questions
Reviewer 1 = R1, Reviewer 2 = R2, Reviewer 3= R3

R1-1 Firstly, the problem of the reflective usage of entrepreneurship exists not only in social entrepreneurship but also in many other emerging fields, including policy entrepreneurship, environmental entrepreneurship, and more recently, religious entrepreneurship. Therefore, I would suggest the authors generally discuss the common themes or problems across these different varieties of entrepreneurship before using social entrepreneurship as a particular example to highlight these common problems. One paragraph at the very beginning of the article will do.”

On page 2 of the revised paper, we have added a paragraph beginning “While in this paper we focus on social entrepreneurship, the spread of this term is part of broader proliferation…” where we now address this issue in some detail.

R1-2 “Secondly, I would suggest the authors discuss the different definitions or perspectives in social entrepreneurship. There have been plenty of reflections on the essential characteristics of social entrepreneurship. Many definitions of social entrepreneurship focus on the innovation and risk-taking of new operative paradigms for social benefits. It is fair to give a mention of these existed efforts. Moreover, for readers who may not be entirely familiar with social entrepreneurship, the authors are obligated to explain and elaborate on the very concept before commenting on its advantages and disadvantages.”

On page  3 of the revised paper, we have added a paragraph beginning “Social entrepreneurship was first used as an expression two decades ago as” where we now address this issue in some detail.

R1-3     “The exact meaning and scope of Catholic social teaching need to be defined more explicitly in the paper. As far as I am aware, there is no official canon of principles or documents associated with Catholic social teaching, even though the phrase is widely used among modern and contemporary Catholic theologists. Each successor of Pope Leo XIII emphasizes slightly different aspects of the term. Therefore, it is inappropriate to assume the concept as given. Instead, the authors need to clarify what they intend to describe or cover by the term. Moreover, Religions is a general journal, so not every reader is familiar with Catholic theology. The authors should also provide some background information on Catholic social teaching to readers who are experts of other religions.”

On pages 4 and 5 of the revised paper, we have added to the paragraph beginning “This essay first lays out two risks we associate with social entrepreneurship…” where we now address this issue in some detail, particularly in the footnote which is attached to this paragraph. We added the details in a footnote so as not to distract from the general flow of the argument for those who are already familiar with the term.

R1-3 “The current conclusion looks a bit too ‘rash’ and ‘thin’ to me. It would be helpful for the authors to summarize their substance and arguments in a more structural and orderly manner.”

On pages 21 and 22 we have completely rewritten the conclusion of the paper. We have attempted to start with a clear summary of our argument as well as clarifying our proposition for the best approach in the future to the term social entrepreneurship. We thank the reviewer as we believe that this change helps greatly to clarify the contribution of the paper.

R1-4 “Beyond these three major issues, I would suggest the authors address a couple of minor issues. For example, there is no need to place a full stop at the end of the title. Also, some paragraphs are very long, and I would suggest the authors break them down where possible.”

We have made the suggested changes on the title page and on pages 7,9,10,11,15 and 17.

Reviewer: 2

Comments:

R2-1 Reviewer 2 did not make any suggested recommendations stating that it was an “[e]excellent paper with a compelling argument--a very good contribution to the field in this area.  I would recommend it for my students.”

Reviewer: 3

Comments:

R3-1 I do think there’s something not quite finished in the article.  The authors argue for placing the common good as the “first principle of entrepreneurship” and argue that will “help us shed light on specific strategies that underpin what is being called social entrepreneurship in a way that enriches the field without degrading the status of more conventional forms of entrepreneur-ship” (p.7).  That’s the move that’s neither finished nor properly developed.  The one-paragraph conclusion of the piece—which follows immediately after the end of the explication of the three “goods”—doesn’t discuss or develop those “specific strategies.” 

On pages 21 and 22 we have completely rewritten the conclusion of the paper. We have attempted to start with a clear summary of our argument as well as clarifying our that our suggestion about the “specific strategies that underpin what is being called social entrepreneurship in a way that enriches the field without degrading the status of more conventional forms of entrepreneur-ship” refers explicitly to the “Three Goods of Entrepreneurship” which we developed in the main body of the paper. It is these three goods which we believe should be definition to all forms of entrepreneurship and the specific requirement of the normative categories are specified in the appeal by drawing on the core concept of Catholic social teaching.  We thank the reviewer as we believe that this change helped us greatly to clarify what we trying to state in the paper. 

R3-2 “I was left very uncertain about what exactly the authors were saying about using the term “social” with respect to entrepreneurship.  One way to read the paper is that they’re saying the term shouldn’t be used at all, if it really is true that “those drawn to the inspiring stories of social entrepreneurship…[should] take this energy and enthusiasm and focus it on exploring the multiple principles of good which can be applied to entrepreneurship in general” (p. 19).  The authors make quite a compelling case that entrepreneurship in general—all entrepreneurship—should be focused on the three goods, and that fits with the page 3 quote I cited earlier, too.  Or do they perhaps mean that the term “social entrepreneurship” should be reserved only for firms that conceive of themselves as attempting to embody the three goods, as opposed to embodying simply the first (as many current “social” entrepreneurial ventures do)?  What is the best “specific strategies” for calling an enterprise “social”?”

The reviewer is correct to assume that what we are suggesting is that “entrepreneurship in general—all entrepreneurship—should be focused on the three goods, and that fits with the page 3 quote I cited earlier, too.” As a result, in our conclusion on pages 22/3, we have added language which specifically states this and hopefully clarifies our position. “We believe, rather than widely adopting the term social entrepreneurship, that we would be better served by directing all the energy and the inspiration provided by the successes of so-called social entrepreneurs to reflect more deeply on our understanding of the term entrepreneurship itself.” This change was most helpful.

Reviewer 2 Report

  1. Key questions about the definition of SE  are right on target.  The dangers of minimizing the impact and worth of non SE are very important. The article is headed in the right direction in my view.
  2. I fear that SE may suffer the same fate as CSR--a faulty premise that business is net value extracting and must give back to balance the scales.
  3. The goods of E are helpful--good goods (as opposed to goods that the market simply demands, nice emphasis here on serving the needs of the poor), good work and good wealth.
  4. Good emphasis on the privatization of character.
  5. Helpful discussion of profit and its role.
  6. Excellent paper with a compelling argument--a very good contribution to the field in this area.  I would recommend it for my students.

Author Response

(The authors gave the same response as above.)

Reviewer 3 Report

Referee Report, February 12, 2021

This piece addresses an important issue: the proper understanding of the term “social entrepreneurship” relative to “entrepreneurship” more generally. 

The context for this paper, the authors argue, is the fact that the boundaries between the two terms have always been difficult to pin down, and that there are both benefits and costs of distinguishing the “social” from the “plain” variety of entrepreneurship.  But ongoing debates on these points have taken attention away from a more fundamental problem, namely, that

the unreflective use of the term social entrepreneurship may perpetuate the idea that “entrepreneurship” is largely a financial and private reality and that this view of entrepreneurship will eventually trivialize and perhaps undermine the important benefits of real intentions behind social entrepreneurship” (p. 2).

They call this point their thesis.  If I was to put this in my own words, I think they mean that the term “entrepreneurship” will come to mean “real,” for-profit entrepreneurship, and by comparison the intentions behind social entrepreneurship will be judged to be minor and non-essential. 

However, I think their true thesis comes in the next full paragraph:

…the widespread problem in defining social entrepreneurship is simply a symptom of a failure to take seriously the central normative core of “entrepreneurship”: that entrepreneurship should have a human end and that this end should be the common good. As a result, attempts to build definitions of social entrepreneurship on purely descriptive contexts and activities risk not only dichotomizing the field into social/moral and non-social/amoral entrepreneurship. More critically, they risk undermining our understanding of the meaning of work itself (p. 3).

So the rest of the paper works out an “organic” approach to thinking about social entrepreneurship, “without relegating entrepreneurship and business in general to the non-social and private sphere” (p. 3).  They root their approach in Catholic Social Teaching, and develop it as a three-legged stool.  They ask entrepreneurs produce “good goods,” organizing the firm so that it embodies “good work” for all its employees—who should be thought of as partners with inherent dignity based on the Imago Dei rather than as merely workers—and “good wealth,” in which the firm distributes profits wisely, justly, and conscientiously (leaning into charity in the deepest sense of the word, caritas).

The bulk of the paper consists of their development of the ideas of good goods, good work, and good wealth.  It’s all quite well-written and well-articulated—and it’s a winsome vision!  I have no quarrel with it or with how they present it.  For Christians, there must be a “central normative core” to entrepreneurship, and the vision for that coming out of CST can be attractive quite broadly to non-Christians and non-religious people as well.

But I do think there’s something not quite finished in the article.  The authors argue for placing the common good as the “first principle of entrepreneurship” and argue that will “help us shed light on specific strategies that underpin what is being called social entrepreneurship in a way that enriches the field without degrading the status of more conventional forms of entrepreneur-ship” (p. 7).  That’s the move that’s neither finished nor properly developed.  The one-paragraph conclusion of the piece—which follows immediately after the end of the explication of the three “goods”—doesn’t discuss or develop those “specific strategies.” 

I was left very uncertain about what exactly the authors were saying about using the term “social” with respect to entrepreneurship.  One way to read the paper is that they’re saying the term shouldn’t be used at all, if it really is true that “those drawn to the inspiring stories of social entrepreneurship…[should] take this energy and enthusiasm and focus it on exploring the multiple principles of good which can be applied to entrepreneurship in general” (p. 19).  The authors make quite a compelling case that entrepreneurship in general—all entrepreneurship—should be focused on the three goods, and that fits with the page 3 quote I cited earlier, too.  Or do they perhaps mean that the term “social entrepreneurship” should be reserved only for firms that conceive of themselves as attempting to embody the three goods, as opposed to embodying simply the first (as many current “social” entrepreneurial ventures do)?  What are the best “specific strategies” for calling an enterprise “social”?

These are the matters this paper must pin down.  I don’t think the revisions need be long—perhaps just a few careful additional paragraphs in the conclusion.  Clarity on these points will make it a much more effective, compelling paper.

Minor point: in the 9th line of the article, “NGOs” not “NGO’s” and for which country do Wong and Tan claim that 7% of national income has been reached?

Author Response

(The authors gave the same response as above.)
